# Flow-Independent Thermal Conductivity and Volumetric Heat Capacity Measurement of Pure Gases and Binary Gas Mixtures Using a Single Heated Wire

**DOI:** 10.3390/mi15060671

**Published:** 2024-05-21

**Authors:** Shirin Azadi Kenari, Remco J. Wiegerink, Remco G. P. Sanders, Joost C. Lötters

**Affiliations:** 1Integrated Devices and Systems, University of Twente, 7522 NB Enschede, The Netherlands; r.j.wiegerink@utwente.nl (R.J.W.); r.g.p.sanders@utwente.nl (R.G.P.S.); j.c.lotters@utwente.nl (J.C.L.); 2Bronkhorst High-Tech BV, 7261 AK Ruurlo, The Netherlands

**Keywords:** thermal flow sensors, thermal conductivity, volumetric heat capacity, ω-3ω technique

## Abstract

Among the different techniques for monitoring the flow rate of various fluids, thermal flow sensors stand out for their straightforward measurement technique. However, the main drawback of these types of sensors is their dependency on the thermal properties of the medium, i.e., thermal conductivity (*k*), and volumetric heat capacity ($\rho c_{p}$). They require calibration whenever the fluid in the system changes. In this paper, we present a single hot wire suspended above a V-groove cavity that is used to measure *k* and $\rho c_{p}$ through DC and AC excitation for both pure gases and binary gas mixtures, respectively. The unique characteristic of the proposed sensor is its independence of the flow velocity, which makes it possible to detect the medium properties while the fluid flows over the sensor chip. The measured error due to fluctuations in flow velocity is less than ±0.5% for all test gases except for He, where it is ±6% due to the limitations of the measurement setup. The working principle and measurement results are discussed.

## 1. Introduction

Thermal flow sensors are used in various application fields, like industry, farming, medicine, etc., where the flow rates of gases needs to be monitored in real-time [1,2]. Thermal flow sensors are cost-effective, have a simple fabrication process, and possess a straightforward working principle. Thermal flow sensors consist of at least one heated element, which is often an electrically heated wire, and the temperature changes due to fluid flow are measured as a function of the flow rate [3,4]. However, the resulting change in temperature is not only dependent on the flow rate; it depends also on the physical properties of the fluid, the thermal conductivity (*k*), and the volumetric heat capacity ($\rho c_{p}$). Therefore, whenever the fluid changes, the sensor needs to be calibrated according to the passing fluid [5,6,7]. Researchers have tried to overcome this problem by implementing different measurement strategies and applying different designs to the sensor or the channel in which the sensor is placed. One of the methods to measure the gas properties *k* and $\rho c_{p}$ is using ω-3ω excitation [8,9,10,11,12,13]. In this technique, an AC current with frequency ω is used for joule heating, resulting in temperature variations at frequency 2ω, and the heater voltage will contain a component at frequency 3ω due to the dependence of the heater’s resistance on temperature. In [14], AC excitation was used to measure the gas properties *k* and $\rho c_{p}$ from the third harmonic voltage amplitude and phase and then correct the flow rate. In [15], the 3ω-method and constant temperature anemometry (CTA) were used to characterize the fluid properties and the flow, respectively. In [16], a thermal flow sensor was used that consisted of a heater and upstream and downstream temperature sensors to determine the flow speed and thermal properties of binary gas mixtures. First, the thermal conductivity is obtained by operating a heater at constant power. Next, the flow speed is obtained using constant temperature operation, correcting the result with the measured thermal conductivity. Finally, the measured thermal conductivity and flow speed are used to obtain the volumetric heat capacity by observing the heater’s temperature decrease at a specific flow speed. By using the thermal properties of the gas mixtures obtained from the previous steps, real-time determination of gas concentration is possible. Other methods have also been presented to obtain the fluid’s properties, such as inserting the sensor in a dead-volume chamber to create a stagnant flow condition [17,18] and using multi-parameter sensors by combining the signals from a Coriolis and thermal flow sensor and a differential pressure sensor to measure different properties such as density, viscosity, specific heat, and thermal conductivity [19,20,21].

Despite all the various methods proposed to overcome the challenges faced by thermal flow sensors, further research on medium-independent flow rate measurement is still required. In this paper, we will show that a single hot wire can measure the gas properties (thermal conductivity and volumetric heat capacity) independent of the flow velocity via different measurement techniques: DC excitation and the 3ω-method, respectively [22].

The paper is organized as follows. In Section 2, the sensor design and working principles are discussed. In Section 3, the fabrication process of the proposed sensor is explained. In Section 4, the measurement results are presented for both thermal conductivity and volumetric heat capacity for a number of gases and binary gas mixtures.

## 2. Design, Theory, and Simulation

The sensor chip contains a single wire suspended over a V-groove cavity, as shown in Figure 1. A V-groove cavity was chosen because it can be easily fabricated using wet etching as will be explained in Section 3. The dimensions of the wire are indicated in Table 1.

The sensor chip is inserted in a tube with circular cross-section such that the wire and V-groove cavity are perpendicular to the flow direction. When the gas enters the tube, it fills the cavity, but the influence of variations in flow velocity on the wire temperature is negligible since the chip is perpendicular to the flow. A COMSOL Multiphysics 6.0 simulation was used to confirm this by simulating the velocity around the suspended wire. Laminar flow physics and either a stationary or time-dependent study are used to simulate nitrogen flow through the channel with the designed sensor inside. Figure 2 shows the velocity magnitude of nitrogen flow inside the channel. It can be seen that the velocity around the sensor wire and close to the channel’s wall is close to zero. Therefore, the only parameters that affect the wire’s temperature are the gas properties. The amount of heat transferred via conduction and convection from the wire to the silicon substrate, which acts as a heat sink, depends on the type of gas due to the different thermal conductivity *k* and volumetric heat capacity $\rho c_{p}$.

To measure the gas physical properties, different measurement techniques, DC and AC excitation, are used to obtain *k* and $\rho c_{p}$, respectively. A schematic illustration of the gas properties measurement is shown in Figure 3.

According to the energy equation, a heated element will transfer heat by three mechanisms: conduction, convection, and radiation. Neglecting the radiation leads to the following energy Equation [14]:(1)$$\rho c_{p}\frac{\delta T}{\delta t}+\rho c_{p}\left(\vec{v}\;\cdot \;\nabla T\right)=k\nabla^{2}T+\dot{q}$$
where $\dot{q}$ is the heat generated in the wire, *T* is the temperature, and *v* is the fluid velocity. According to the energy equation, the amount of heat transfer through a heated object is dependent on the thermal conductivity and volumetric heat capacity of the surrounding fluid, which can be detected by applying different excitations:With DC excitation under stagnant flow conditions, the two terms on the left side of the energy equation can be ignored, resulting in $\dot{q}=-k\nabla^{2}T$, which is only dependent on *k*.With AC excitation and stagnant flow conditions, only the second term on the left side of the energy equation can be neglected. Then, the equation becomes $\rho c_{p}\frac{\delta T}{\delta t}=k\nabla^{2}T+\dot{q}$, which is dependent on both *k* and $\rho c_{p}$.

### 2.1. Thermal Conductivity

Figure 4a shows the simplified lumped-element equivalent circuit in the thermal domain for the suspended wire over a V-groove cavity when it is heated by a DC current. The temperature of the wire is the product of heating power *P* and thermal resistance $R_{th}$ through the fluid between the wire and the silicon substrate (inside the V-groove cavity). $R_{th}$ depends on the type of gas, as different gases have different thermal conductivities. Assuming that the heating power is equally distributed over the length of the wire, the temperature distribution over the length of the wire can be calculated [23]:(2)$$\frac{T(x_{n})}{P^{\prime }}=\frac{1}{G_{f}^{\prime }}\left(1-\frac{\cosh(x_{n}l\sqrt{R_{b}^{\prime }\;G_{f}^{\prime }})}{\cosh(0.5l\sqrt{R_{b}^{\prime }\;G_{f}^{\prime }})}\right)$$
where $x_{n}$ is the normalized position on the wire, $P^{\prime }$ is the electrical line power in [W/m], $G_{f}^{\prime }$ is the line conductance through the gas in [W/(K·m)], and $R_{b}^{\prime }$ is the thermal line resistance of the beam in [K/(W·m)].

Figure 5a shows the calculated temperature distribution over the length of the wire for different pure and binary mixtures of gases. From the large region with constant temperature, it can be concluded that the wire temperature is mainly defined by conduction through the gas. Only near the ends of the wire, there is an effect of conduction through the wire itself. By integrating the temperature distribution over the wire length, the average temperature can be obtained:(3)$$T_{ave}=\frac{1}{l}\int_{0}^{l}\Delta T\left(x\right)dx$$

The average temperature is used to calculate the electrical resistance of the wire when heated, $R_{h}$, and the resulting voltage drop over the wire for a heating current $I_{dc}$:(4)$$V=R_{h}I=R_{a}\left(1+\alpha T_{ave}\right)I_{dc}$$
where $R_{a}$ is the electrical resistance of the wire at room temperature and α is the temperature coefficient of resistance. Figure 5b shows the calculated voltage drop as a function of 1/k for a heating current of 2 mA. It can be seen that the average temperature of the wire increases approximately linearly with 1/k.

### 2.2. Volumetric Heat Capacity

The lumped-element equivalent circuit for the suspended wire in case of AC heating is shown in Figure 4b. The temperature fluctuations now depend on the thermal capacitance as well as the thermal resistance. By applying an AC current to the wire, the wire will heat up at a frequency ω. It will produce temperature and wire resistance variations at a frequency of 2ω, resulting in a frequency component in voltage at 3ω. This third harmonic contains information about the thermal conductivity and volumetric heat capacity of the fluid. Figure 6 illustrates the operation principle of the 3ω-method. The voltage over the heated wire due to excitation with an AC current $i=I_{ac}$ cos(ωt), with $I_{ac}$ the amplitude of the current, can be expressed as follows [22]:
(5)$$\begin{array}{cc} V(t) & =R\;\cdot \;i=R\;\cdot \;I_{ac}\cos\left(\omega t\right) \end{array}$$
with the resistance
(6)$$R=R_{a}\left(1+\alpha \Delta T\right)$$
being dependent on the resulting temperature changes ΔT. The heating power can be expressed as:(7)$$\begin{array}{cc} P=R_{h}\;\cdot \;i^{2}=\frac{R_{h}I_{ac}^{2}}{2}\left(1+\cos\left(2\omega t\right)\right) & =\frac{R_{h}I_{ac}^{2}}{2}+\frac{R_{h}I_{ac}^{2}}{2}\cos\left(2\omega t\right)\\ \; & =P_{dc}+P_{ac}\cos\left(2\omega t\right) \end{array}$$
with $R_{h}$ the electrical resistance of the heated wire due to the average heating power $P_{dc}$. Here, we neglect the AC changes in resistance at frequency 2ω since these are relatively small. Using the equivalent circuit in Figure 4b, we find the following for the resulting temperature changes:(8)$$\begin{array}{cc} \Delta T & =P_{dc}|Z_{dc}|+P_{ac}\left|Z_{2\omega }\right|\cos\left(2\omega t+∠Z_{2\omega }\right)\\ \; & =\frac{R_{h}\;\cdot \;I_{ac}^{2}}{2}R_{th}-\frac{R_{h}\;\cdot \;I_{ac}^{2}}{2}\left|Z_{2\omega }\right|\cos\left(2\omega t+∠Z_{2\omega }\right)\\ \; & =\Delta T_{dc}+\Delta T_{ac}\cos\left(2\omega t+∠Z_{2\omega }\right) \end{array}$$

Inserting (Equation 8) into (Equation 5) we find for the voltage drop over the wire:(9)$$\begin{array}{cc} V(t) & =I_{ac}R_{a}\left(1+\alpha \Delta T_{dc}\right)\cos\left(\omega t\right)\\ \; & +\frac{I_{ac}R_{a}\alpha \Delta T_{ac}}{2}\cos\left(\omega t+∠Z_{2\omega }\right)\\ \; & +\frac{I_{ac}R_{a}\alpha \Delta T_{ac}}{2}\cos\left(3\omega t+∠Z_{2\omega }\right) \end{array}$$

The voltage has two time-dependent terms at the frequencies of ω and 3ω. The amplitude of the 3ω component is highly dependent on the fluid properties and can be described as follows:(10)$$|V_{3\omega }|=\frac{I_{ac}^{3}R_{h}R_{a}\alpha }{4}\left|Z_{2\omega }\right|$$

In the amplitude of the third harmonic, two parameters play a role ($R_{th}=\frac{l}{kA}$ and $C_{th}=\rho c_{p}V$) since the impedance (*Z*) is a function of $R_{th}$ and $C_{th}$ which relate to *k* and $\rho c_{p}$, respectively. Here, *V* is the effective volume of the V-groove cavity that defines how much fluid is contributing to the thermal capacitance, and *A* and *l* are the effective cross-sectional area and depth of the V-groove that define the thermal resistance towards the silicon substrate. A COMSOL simulation is employed to obtain the amplitude of the third harmonic $|V_{3\omega }|$ for the wire fed with an arbitrary current. Laminar flow, heat transfer in solids and fluids, and electric currents physics are used. A sine waveform with an amplitude of 2 mA and a frequency of 4 kHz is applied to the wire and a time-dependent study is used to obtain the voltage over the wire as a function of time, as shown in Figure 7a. The voltage contains 1ω and 3ω components. To calculate the amplitude of the third harmonic, a fast Fourier transform (FFT) is implemented. The calculated $|V_{3\omega }|$ is plotted in Figure 7b as a function of the product of *k* and $\rho c_{p}$ for four different gases, Ar, CO_2_, air, and He. $|V_{3\omega }|$ decreases approximately linearly with $k\rho c_{p}$.

## 3. Fabrication

Figure 8 shows a summary of the fabrication process. To fabricate the sensor, first a layer of 300 nm silicon rich silicon nitride (SiRN) is deposited on a standard (100) silicon wafer by low-pressure chemical vapor deposition (LPCVD), (Figure 8a). Then, a 10 nm Cr adhesion layer and a 100 nm Pt layer are deposited and etched by sputtering and ion beam etching (IBE), respectively, to pattern the wires and metal tracks (Figure 8b,c). The IBE step is repeated with the mask that defines the V-groove cavity (Figure 8d). This step narrows down the width of the sensor wire and provides self-alignment with the cavity. Next, in Figure 8e, the SiRN layer is etched by plasma etching to open the window for etching the silicon. Finally, the silicon substrate is etched in a KOH solution (KOH 1:3 DI-water) to realize the V-groove cavity, (Figure 8f). Figure 9 shows two microscope pictures of a fabricated chip.

## 4. Results and Discussion

Figure 10 shows a schematic drawing of the measurement setup. There are two flow controllers (μ-Coriolis flow controller ML120V21, Bronkhorst, Ruurlo, The Netherlands) and pressure controllers (M19205673D, Bronkhorst, Ruurlo, The Netherlands) to apply the pure gases (CO_2_, Ar, Air, He) and binary gas mixtures (Ar–Air and Ar–He) into the sensor tube. Each gas is applied to the system separately in four different flow rates from 10 g/h to 40 g/h. A minimum flow rate of 10 g/h was used so that the gas does not mix with outside air.

To measure the thermal conductivity, a 2 mA DC current is applied to the sensor wire using an HP3245A Universal Source (Hewlett-Packard Company, USA), and the voltage drop over the wire is measured with a Keysight DAQ970A Data Acquisition System (Keysight Technologies, USA). Figure 11a shows the measured voltage as a function of flow rate. As can be seen, the voltage drop over the wire is almost independent of the flow rate. For each gas, the maximum error due to flow is calculated using:(11)$$max\;error\%=\frac{max\;voltage\;value-mean\;voltage\;value}{mean\;voltage\;value}\times 100$$

An overview of the calculated errors is shown in Table 2. The maximum error for He is much larger than for the other gases, most likely because in the setup that is used, He mixes easily with air entering from the tube outlet, which is most noticeable at low flow rates. Figure 11b shows the voltage drop as a function of 1/k of the gas. The measured voltages, theoretical values, and simulation results are compared together. The vertical blue error bars indicate the voltage variations due to flow velocity changes for flows up to 40 g/h. These variations are small, showing that the sensor is independent of the velocity. The horizontal red error bars correspond to the variation in *k* over the temperature range from $20{\;}^{\circ }$C to $150{\;}^{\circ }$C. The thermal conductivity is highly dependent on the temperature. For the plot in Figure 11b, it is assumed that the temperature of the gas is $80{\;}^{\circ }$C and the corresponding thermal conductivity is obtained from FLUIDAT at a pressure of 1 bar [24].

For the measurement of volumetric heat capacity, $\rho c_{p}$, an AC current of 4 mA peak to peak is applied to the sensor wire, and the voltage amplitude of the third harmonic $|V_{3\omega }|$ is measured with a Stanford Research Systems SR830 lock-in amplifier. Figure 12a shows the measured amplitude $|V_{3\omega }|$ as a function of frequency. The frequency of the 3ω component must be higher than the thermal cut-off frequency to contain the gas’s volumetric heat capacity ($\rho c_{p}$) as well as thermal conductivity (*k*) information. Therefore, an actuation frequency of 4 kHz was chosen, resulting in a 3ω component at 12 kHz. Figure 12b shows the measured amplitude of the 3ω component as a function of flow for different gases. The same procedure as for the DC measurement was applied to calculate the maximum error due to velocity fluctuations. The maximum error is calculated with Equation (Equation 11), and the obtained values are shown in Table 2. Again, the maximum error is observed for He (due to the mixing with air at low flow rates) which is less than 6%.

Figure 13 shows the measured amplitude of the 3ω component for different gases and three different actuation frequencies as a function of $k\rho c_{p}$. For the actuation frequency of 4 kHz, the temperature of the wire is also simulated in COMSOL for four different gases, and the amplitude of the 3ω component is calculated with a fast Fourier transform (FFT). The vertical blue error bars indicate the fluctuations due to the flow velocity. The horizontal red error bars show the effect of the dependence of *k* on temperature, for a temperature range from 20 to $150{\;}^{\circ }$C.

The measurement results show that a single wire suspended over a V-groove cavity can measure the thermal conductivity and volumetric heat capacity of both pure gases and binary mixtures of gases through different types of excitation since the wire is only dependent on the medium, not the fluid velocity. The dependency on the velocity is eliminated by positioning the wire perpendicular to the flow direction, which makes the velocity around the wire stagnant. However, the velocity fluctuations for gases lighter in comparison to air like He are still high. To improve the sensor’s results and minimize the error, the 3D-printed tube design should be improved to reduce the leakage.

## 5. Conclusions

Various gases and binary gas mixtures (CO_2_, Ar, Air, He, Ar–He, Air–Ar) have been applied to a single heated wire suspended above a V-groove cavity, and the fluid’s thermal conductivity and volumetric heat capacity have been measured. To detect the thermal conductivity, the wire is heated by a DC current and the voltage drop over the wire is measured. To detect the volumetric heat capacity, the wire is heated by an AC current and the amplitude of the third harmonic in the voltage is measured. The measurement results have been compared to both theory and simulation, showing good agreement. It is shown that a single wire can be used to measure the thermal properties of gases independent of the flow velocity while the flow is changing. The obtained maximum error due to changes in flow velocity is less than 0.5%, except for He, for which it is less than 6%. The significantly larger error for He is most likely due to the measurement setup, in which helium can easily mix with air entering from the outlet, which is most noticeable at low flow velocities. The thermal conductivity and volumetric heat capacity ranges used in this paper are from 0.01652 to 0.1554 $\frac{W}{m\;\cdot \;K}$ and 864 to 1558 $\frac{J}{K\;\cdot \;m^{3}}$, respectively.

Future research includes solving the problem for He, doing additional measurements with other types of gases, and using the physical parameters, thermal conductivity and volumetric heat capacity of pure and binary mixtures of different gases to correct the flow rate of the proposed calorimetric flow sensor in [23]. 

## Figures and Tables

**Figure 1 micromachines-15-00671-f001:**
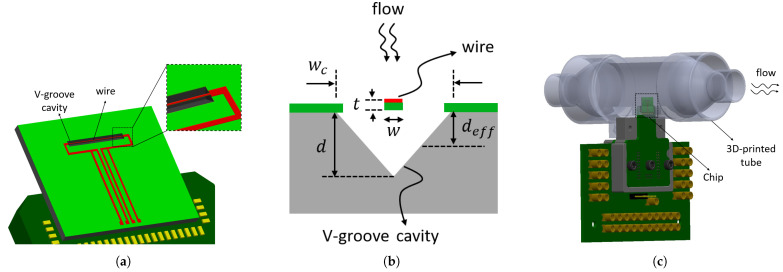
Schematic drawing of the single wire suspended on a V-groove cavity. (**a**) Top view, (**b**) cross-section view of the sensor ($d_{eff}\approx \frac{d}{2}$), and (**c**) chip inserted in a 3D-printed tube.

**Figure 2 micromachines-15-00671-f002:**
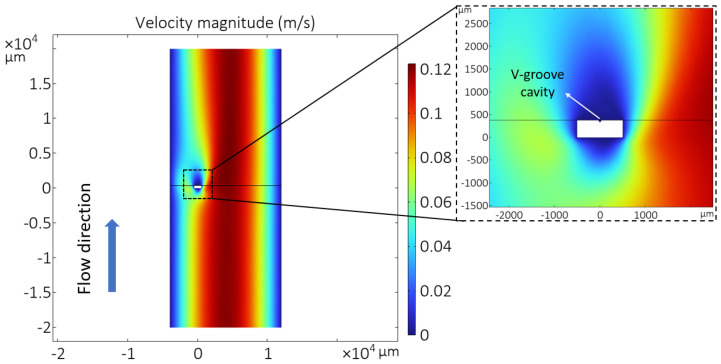
COMSOL simulation results of the wire suspended on a V-groove cavity (see Figure 1b) indicating that the velocity around the sensor is negligible.

**Figure 3 micromachines-15-00671-f003:**
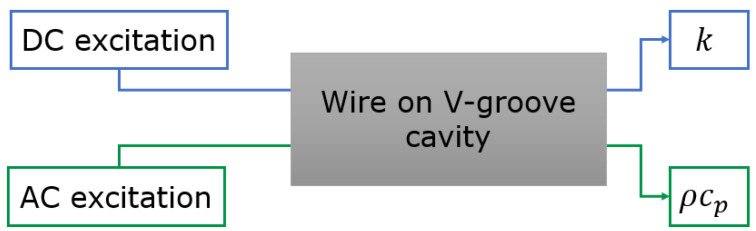
General schematic of the gas physical property measurement method.

**Figure 4 micromachines-15-00671-f004:**
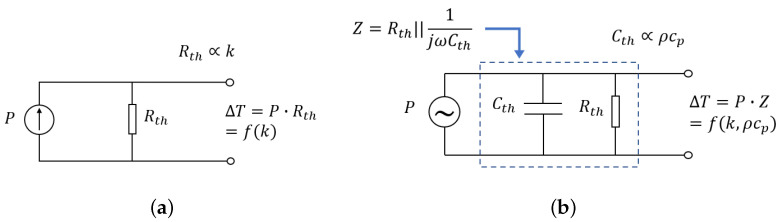
Equivalent circuit for a wire suspended on a V-groove cavity fed by (**a**) a DC and (**b**) an AC current. $R_{th}$ and $C_{th}$ are the thermal resistance and thermal capacitance of the fluid.

**Figure 5 micromachines-15-00671-f005:**
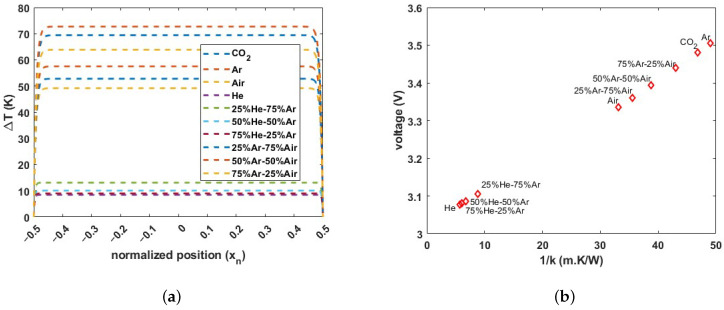
(**a**) Calculated temperature distribution over the length of the wire ($\Delta T=T_{wire}-T_{ambient}$) using Equation (Equation 2) for pure and binary mixtures of gases at a heating current of 2 mA and (**b**) the calculated voltage drop over the wire using Equation (Equation 4) plotted as a function of 1/k of the gas.

**Figure 6 micromachines-15-00671-f006:**
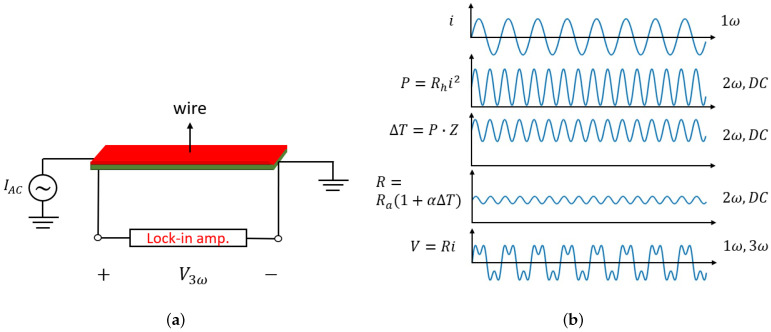
General description of the 3ω-method, (**a**) the sensor wire is heated with an AC current and the amplitude of the third harmonic is measured with a lock-in amplifier, (**b**) schematic illustration of the relationship between the sinusoidal current, power, temperature, resistance, and the voltage drop over the wire.

**Figure 7 micromachines-15-00671-f007:**
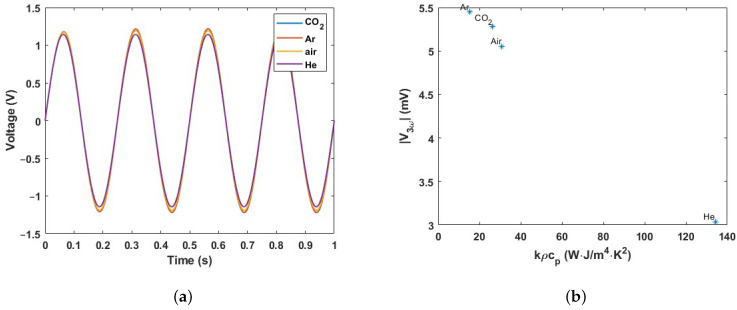
(**a**) Simulated voltage over the wire for different pure gases plotted as a function of time, and (**b**) amplitude of the third harmonic $|V_{3\omega }|$ obtained by applying a fast Fourier transform (FFT) plotted as a function of $k\rho c_{p}$.

**Figure 8 micromachines-15-00671-f008:**
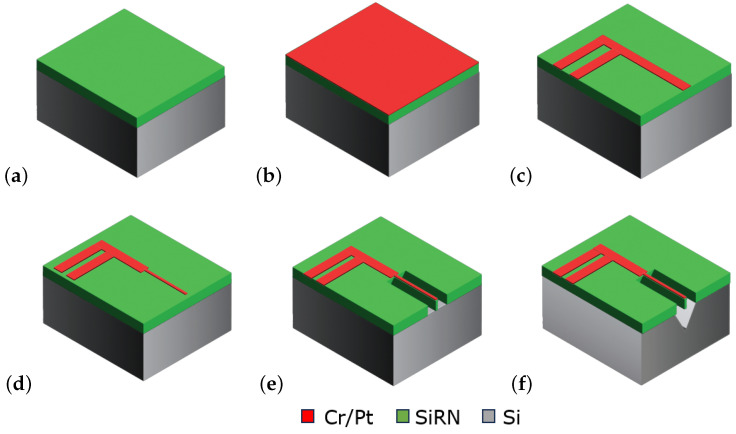
Schematic drawing of the fabrication process flow. (**a**) SiRN deposition by LPCVD, (**b**) Cr/Pt sputtering, (**c**) pattern the metal traces via lithography and IBE etch of the uncovered metals, (**d**) second lithography to make the metal layer narrower and open the window on SiRN for wet etching of the silicon, and IBE etch of the metal, (**e**) etch the SiRN layer with RIE, and (**f**) immerse the wafer into a KOH vessel for etching the silicon to make the V-groove cavity.

**Figure 9 micromachines-15-00671-f009:**
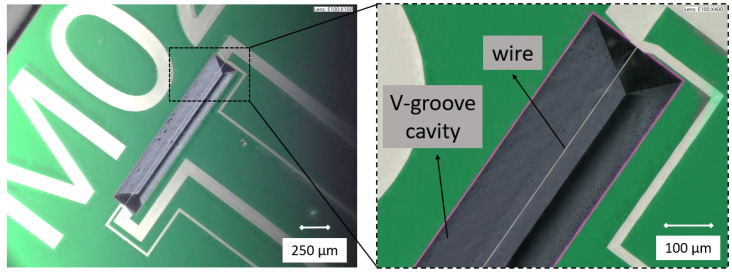
Microscope photograph of the released sensor.

**Figure 10 micromachines-15-00671-f010:**
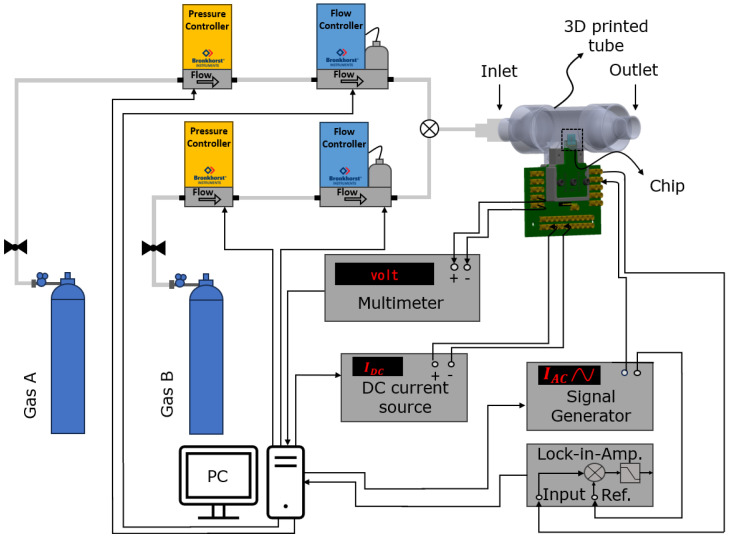
Schematic drawing of the measurement setup.

**Figure 11 micromachines-15-00671-f011:**
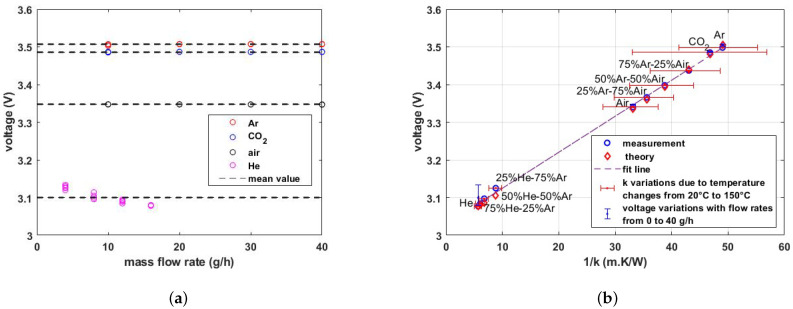
Measured voltage drop over the wire with a DC excitation current of 2 mA. (**a**) Plotted as a function of flow rate, showing little influence of the flow velocity except for helium. (**b**) Plotted as a function of 1/k for various pure and binary mixtures of gases, showing a clear dependence on the thermal conductivity of the gas.

**Figure 12 micromachines-15-00671-f012:**
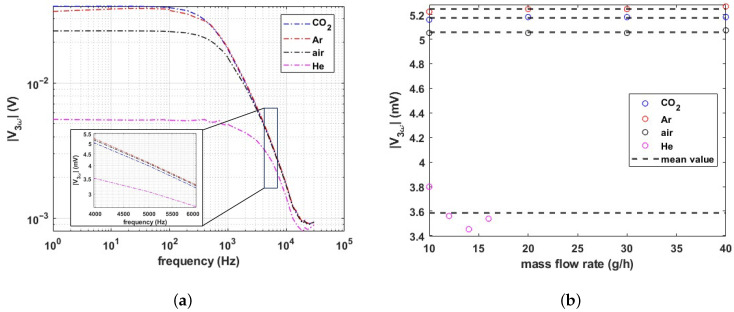
Measured amplitude of the third harmonic in the voltage drop over the wire in case of AC excitation with a sinusoidal current with an amplitude of 2 mA. (**a**) Frequency sweep from 1 Hz to 30 kHz. The inset shows the response around 4 kHz, which is used for measurement of the volumetric heat capacitance. (**b**) Measured 3ω amplitude at the excitation frequency of 4 kHz as a function of flow rate.

**Figure 13 micromachines-15-00671-f013:**
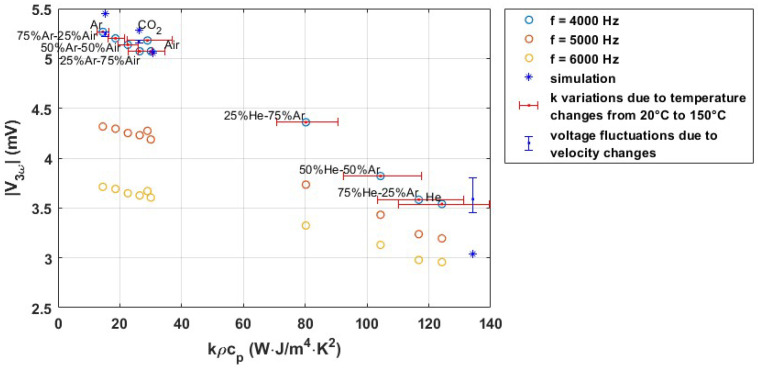
Measured and simulated 3ω amplitude for the sensor wire with AC excitation, with a sinusoidal current at different frequencies and an amplitude of 2 mA, plotted as a function of $k\rho c_{p}$ for various pure and binary mixtures of gases.

**Table 1 micromachines-15-00671-t001:** The dimensions of the wire.

Parameter	Symbol	Value
Beam length	*l*	2 mm
Beam width	*w*	3 μm
Beam thickness	*t*	400 nm
V-groove width	$w_{c}$	80 μm
V-groove depth	*d*	58 μm

**Table 2 micromachines-15-00671-t002:** The error percentage of the voltage fluctuations due to the changes in the velocity.

Fluid		CO_2_	Ar	Air	He
Max error (%)	DC	<0.03	<0.04	<0.008	<1.5
	AC	<0.3	<0.4	<0.3	<6

## Data Availability

Data are contained within the article.

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
