# Peer review of "Flow-Independent Thermal Conductivity and Volumetric Heat Capacity Measurement of Pure Gases and Binary Gas Mixtures Using a Single Heated Wire"

_micromachines, 2024, doi:10.3390/mi15060671_

Round 1

Reviewer 1 Report

Comments and Suggestions for Authors

The manuscripta designed a structure of "single hot wire suspended above a V-groove cavity" to measure tempearture change of gas flows. The idea is creative and practical. The manuscript is well written and the results are exhibited.

1. Figure 5 a missed unit?

2. Air itself is already a mixture, could it have some influence on the results, why not use N2 or O2 seperately? They are not expensive too.

3. How about it is used to measure H2?

Author Response

The manuscripta designed a structure of "single hot wire suspended above a V-groove cavity" to measure tempearture change of gas flows. The idea is creative and practical. The manuscript is well written and the results are exhibited.

  1. Figure 5 a missed unit?

Thank you very much for your comment. We now added the unit for ΔT.

  1. Air itself is already a mixture, could it have some influence on the results, why not use N2 or O2 seperately? They are not expensive too.

Indeed, it is possible to use N2, as we did in our previous paper [1] about only the thermal conductivity measurement. Using pure oxygen is not allowed in our labs. This time we used air, because the difference with N2 is small.

  1. How about it is used to measure H2?

The sensor could also be used with H2, however we do not have this gas in our lab.

References

[1] S. Azadi Kenari, R. J. Wiegerink, R.G.P. Sanders, J.C. Lötters, ``Thermal Flow Meter with Integrated Thermal Conductivity Sensor'', micromachines 14, 2023.

Reviewer 2 Report

Comments and Suggestions for Authors

Authors have introduced an innovative technique to measure thermal conductivity and volumetric heat capacity using a single wire heated by DC and AC currents, respectively. The study demonstrates that the measurement results are independent of the flow rate with the exception of He gas.

While the manuscript is clearly written and outlines a promising method, it is notable that the application of this technique in thermal flow sensors, central to the study's aim, was not tested to assess potential improvements in measurement accuracy of the sensor itself. Additional comments can be found below:

1. Could you explain the choice of a V-shaped groove in your experimental setup?

2. Would the positioning of the wire within the system influence the results?

3. What was the rationale behind selecting the range of 10 g/h to 40 g/h for the flow rate?

Author Response

Authors have introduced an innovative technique to measure thermal conductivity and volumetric heat capacity using a single wire heated by DC and AC currents, respectively. The study demonstrates that the measurement results are independent of the flow rate with the exception of He gas.

While the manuscript is clearly written and outlines a promising method, it is notable that the application of this technique in thermal flow sensors, central to the study's aim, was not tested to assess potential improvements in measurement accuracy of the sensor itself.

Indeed, the current paper presents only the measurement of thermal conductivity and volumetric heat capacity of the gas. Initial results on using this sensor in combination with a thermal flow sensor were presented at the last MEMS conference and will be published in a separate paper.

Additional comments can be found below:

  1. Could you explain the choice of a V-shaped groove in your experimental setup?

Thank you very much for your comment. We chose a V-groove because it can be easily fabricated using KOH etching. We have now also added this in the paper (line 58).

  1. Would the positioning of the wire within the system influence the results?

Definitely yes. The wire needs to be perpendicular to the flow direction to eliminate the flow fluctuations’ influence on the output signal, subsequently it only depends on the gas properties. In the paper this is described in lines 59 to 62.

  1. What was the rationale behind selecting the range of 10 g/h to 40 g/h for the flow rate?

The minimum of 10 g/h is chosen due to the setup limitations. It is chosen to be sure that the tube is filled with gas, and no air, entering from the outlet of the tube, is mixed to the gas.

For gases with higher density, like CO2, with the pressure of 8 bar, the maximum flow rate that the flow controller can provide is almost 40 g/h. Therefore, this maximum limit is chosen for the other gases to have the same range for the rest.

We added a remark in lines 162 and 163, and we also modified the figures such that the flow rate is always given in g/h.